# Hyperaccumulator *Stanleya pinnata*: In Situ Fitness in Relation to Tissue Selenium Concentration

**DOI:** 10.3390/plants11050690

**Published:** 2022-03-03

**Authors:** Leonardo Warzea Lima, McKenna Castleberry, Ami L. Wangeline, Bernadette Aguirre, Stefano Dall’Acqua, Elizabeth A. H. Pilon-Smits, Michela Schiavon

**Affiliations:** 1Biology Department, Colorado State University, Fort Collins, CO 80523, USA; leolima@rams.colostate.edu (L.W.L.); kenna.castleberry@gmail.com (M.C.); epsmits@colostate.edu (E.A.H.P.-S.); 2Biology Department, Laramie County Community College, Cheyenne, WY 82007, USA; awangeli@lccc.wy.edu (A.L.W.); bernadetteaguirre82@gmail.com (B.A.); 3Department of Pharmaceutical and Pharmacological Sciences, University of Padova, Via Marzolo 5, 35131 Padova, Italy; stefano.dallacqua@unipd.it; 4Dipartimento di Scienze Agrarie, Forestali e Alimentari, University of Torino, Largo Paolo Braccini 2, 10095 Grugliasco, Italy

**Keywords:** selenium, herbivory, glucosinolates, flavonoids, plant fitness, seleniferous soil

## Abstract

Earlier studies have shown that *Stanleya pinnata* benefits from selenium hyperaccumulation through ecological benefits and enhanced growth. However, no investigation has assayed the effects of Se hyperaccumulation on plant fitness in the field. This research aimed to analyze how variation in Se accumulation affects *S. pinnata* fitness, judged from physiological and biochemical performance parameters and herbivory while growing naturally on two seleniferous sites. Natural variation in Se concentration in vegetative and reproductive tissues was determined, and correlations were explored between Se levels with fitness parameters, herbivory damage, and plant defense compounds. Leaf Se concentration varied between 13- and 55-fold in the two populations, averaging 868 and 2482 mg kg^−1^ dry weight (DW). Furthermore, 83% and 31% of plants from the two populations showed Se hyperaccumulator levels in leaves (>1000 mg kg^−1^ DW). In seeds, the Se levels varied 3–4-fold and averaged 3372 and 2267 mg kg^−1^ DW, well above the hyperaccumulator threshold. Plant size and reproductive parameters were not correlated with Se concentration. There was significant herbivory pressure even on the highest-Se plants, likely from Se-resistant herbivores. We conclude that the variation in Se hyperaccumulation did not appear to enhance or compromise *S. pinnata* fitness in seleniferous habitats within the observed Se range.

## 1. Introduction

Selenium (Se) is recognized as an indispensable nutrient for many animals, prokaryotes, and microalgae [1], functioning as a structural component of selenoproteins. The essentiality of Se to higher plants is not yet verified [1]; instead, this element is considered beneficial [2]. Selenium can induce the cellular antioxidant system at low levels [3], with a variety of advantageous responses, such as enhanced growth, more efficient photosynthesis, higher accumulation of starch and sugars, delayed senescence, and protection against oxidative stress [3,4,5]. Nonetheless, the threshold between Se adequacy and toxicity is very narrow for many species. Most plants, for instance, experience toxicity above 100 μg g^−1^ DW (dry weight) and are defined as Se-non accumulators.

Plants likely lost their Se-specific metabolism during evolution, since no molecular mechanisms have been discovered that specifically insert seleno-amino acids into proteins [6,7]. In plant cells, Se can be assimilated nonspecifically into the Se-amino acids selenocysteine (SeCys) and selenomethionine (SeMet) by accessing the assimilation pathway of its analog sulfur (S) [8]. The insertion of Se amino acids into proteins in place of the S amino acids cysteine (Cys) and methionine (Met) can produce malformed proteins that lose their function, and this has been established as a major cause of Se toxicity to plants [9,10]. In addition, inorganic Se can cause oxidative stress at higher tissue concentrations in most species [10].

The uptake of Se by roots tightly depends on soil Se concentration, phytoavailability, and Se speciation [11]. The average Se level in the soil is usually below 2 μg g^−1^ but can reach up to 1.2 mg g^−1^ in soils derived from seleniferous sedimentary rocks [12]. These areas can be found in the great plains of the United States of America, including the seleniferous shale formations at Coyote Ridge and Pine Ridge Natural Areas in Colorado [13,14]. Some plant taxa growing in these naturally seleniferous areas can efficiently accumulate Se in their leaf tissues but display different physiological and ecological strategies to cope with the high Se concentration. These plants are classified into two major categories, namely accumulators (or secondary accumulators), which can concentrate from 100 to 1000 μg Se g^−1^ DW, and hyperaccumulators, which can exhibit 1000–10,000 μg Se g^−1^ DW (up to 0.1% of Se per DW) [15].

To date, more than 500 plant species have been described to hyperaccumulate a specific non-essential element [16]. In the previous 15 years, much knowledge has been gained on Se hyperaccumulator plant species, and, to date, this trait is reported in different families in the orders Malpighiales, Brassicales, and Asterales [17,18], forming a group with no common ancestor [18], where the Brassicaceae constitutes the most represented family, with more than 100 taxa [17,18]. The Se-hyperaccumulator *Stanleya pinnata* (Brassicaceae), also known as Prince’s plume, is a desert perennial plant native to most of the arid western part of the United States, including the great plains, and can be found throughout the state of Colorado [17,19]. This particular plant can hyperaccumulate Se up to 0.1% of its dry weight, mainly as organic and less toxic forms of Se [17]. Transcriptomic studies have revealed that *S. pinnata* possesses an elevated expression of several genes that have a role in plant defense against stress, either abiotic or biotic [19,20]. The overexpression of such genes corresponds with alternative biochemical mechanisms evolved by the Se-hyperaccumulator that efficiently detoxify or remove excessive Se via methylation of SeCys and subsequent volatilization, thus preventing Se misincorporation into proteins [1,21].

Boyd and Martens [22] described the elemental defense hypothesis, stating that the hyperaccumulation trait likely evolved because it confers certain ecological advantages. High levels of Se in plants, in particular, can be toxic to herbivores. Thus, by accumulating extremely high Se levels in their tissues, *S. pinnata* plants can have greater protection from pathogens or herbivore attacks [23,24,25]. Plant Se accumulation can offer protection against herbivory through either deterrence or toxicity [26,27,28]. Various invertebrate and vertebrate herbivores were shown to actively avoid plants treated with Se and suffered toxicity when forced to feed on high-Se plants [29]. Field surveys also showed a possible protective effect of high Se plants against herbivory: Se hyperaccumulator species growing in the seleniferous Pine Ridge Natural Area sheltered fewer arthropods when compared to non-accumulators [30]. Se hyperaccumulator plants may also function in elemental allelopathy against non-accumulator neighboring plants; indeed, by accumulating hyperaccumulator Se levels in their tissues and depositing this element in the litter, *S. pinnata* plants can better compete with the surrounding Se-sensitive vegetation [1,27,28,29,31].

While Se protects plants against many generalist herbivores, Se-resistant leaf and seed herbivores were found to live in symbiosis with hyperaccumulators *S. pinnata* and *Astragalus bisulcatus*. These herbivores can utilize high-Se plants as a food source without experiencing toxicity, either via Se exclusion or Se tolerance [32,33,34,35]. For example, Freeman et al. [32] found that a Colorado population of the diamondback moth (*Plutella xylostella*) was able to tolerate high tissue levels of Se by avoiding demethylation of the plant’s primary form of Se, methyl-SeCys, thus avoiding its incorporation into proteins, in contrast to another population from a non-seleniferous area that was not. In another study, Freeman et al. [33] found a parasitoid chalcid wasp in the seeds of *S. pinnata* that resisted Se via exclusion. Similarly, Valdez Barillas et al. [34] found two different herbivore moth species on *A. bisulcatus*. Various other types of symbionts of hyperaccumulators also appear to have co-evolved with them by developing Se resistance: litter decomposers [24,36], pollinators [37], endophytic, and rhizosphere microbes [38,39]. There is also evidence that hyperaccumulators may facilitate Se-tolerant plant species, which were often found growing near hyperaccumulators, where they also had elevated Se levels and suffered less herbivory damage [27,28].

Thus, the emerging picture from earlier studies is that Se hyperaccumulator plant species can positively and negatively affect different ecological partners, depending on these partners’ Se resistance. Through these combined effects, hyperaccumulators may affect the local species composition of animals [32,33], plants [13,14,27,28], mycorrhizal, rhizosphere, and endophytic microorganisms [38,40].

From collective earlier physiological, biochemical, and ecological studies, it is clear that *S. pinnata* benefits in several ways from Se accumulation through the described ecological benefits and enhanced growth [29,37]. So far, there is no evidence for any physiological or ecological constraints. However, no investigation has yet been reported on the effects of Se hyperaccumulation on the overall plant fitness in the field. To investigate to what extent Se hyperaccumulation contributes to *S. pinnata* fitness, a field survey was conducted within two wild populations growing in naturally seleniferous areas. Variation in Se concentration in vegetative and reproductive tissues was determined, and correlations were explored between the observed Se levels with fitness parameters, herbivory damage, and classes of biochemical defense compounds. It is hypothesized that plant Se concentration positively correlates with the various fitness and physiological parameters studied. However, this correlation curve would likely be most pronounced at relatively lower Se tissue levels, perhaps saturating at a certain tissue Se threshold. An inverse correlation between Se concentration and herbivory is predicted.

## 2. Results

### 2.1. Selenium Accumulation

As a first step to investigate how Se accumulation affects *S. pinnata* fitness, the degree of variation in Se accumulation was surveyed. Two wild populations were sampled, growing in their native habitat on a naturally seleniferous shale formation at Coyote Ridge Natural Area and Pine Ridge Natural Area near Fort Collins, Colorado, USA (Figure 1). Both populations have been studied extensively [13,14,27,28,33], but not concerning fitness parameters in the field. The Coyote Ridge population was sampled in spring, and the Pine Ridge population was sampled in the fall. Substantial variation in leaf Se concentration was found within each population. Leaf Se concentration varied 55-fold among the 23 sampled Coyote Ridge plants (Figure 2A) and 13-fold among the 24 sampled Pine Ridge plants (Figure 3A). Overall, the leaf Se concentration was higher in the Pine Ridge plants, which showed an average leaf Se concentration of 2482 mg kg^−1^ dry weight (DW), while the Coyote Ridge plants showed an average of 868 mg kg^−1^ DW. Furthermore, 83% of the sampled Pine Ridge plants showed Se hyperaccumulator levels in their leaves (>1000 mg kg^−1^ DW) versus 31% of the Coyote Ridge plants.

The Se variation was more pronounced in the siliques when compared to the seeds in all plants analyzed from both sites. Only 4-fold (Figure 2B) and 3-fold (Figure 3B) seed Se variations were found, respectively, among Coyote Ridge and Pine Ridge plants. A greater Se variation, 130-fold, was found in the siliques from plants sampled in Coyote Ridge (Figure 2B), and a 4-fold variation was determined in the Pine Ridge group (Figure 3B). The large silique Se variation in Coyote Ridge reflected the relatively low Se levels found in a few sampled plants; indeed, 17% of all plants from that site showed Se levels below 750 mg kg^−1^ DW.

All the seeds and most of the siliques analyzed contained Se concentrations typical of Se hyperaccumulators for both studied sites. On average, the plants collected from Coyote Ridge accumulated 4043 mg kg^−1^ DW and 2267 mg kg^−1^ DW in seeds and siliques, respectively. The Pine Ridge plants contained on average 3372 mg kg^−1^ DW in seeds and 3323 mg kg^−1^ DW in the siliques. It should be noted that the seed Se concentration was consistently high for all plants analyzed, even when the silique Se level was low in the same plant.

On average, the Se concentration was higher in the reproductive organs than the leaf tissues for the Coyote Ridge plants. A 3-fold higher Se concentration in the siliques and 5-fold higher Se in seeds were found compared to the leaves. The Pine Ridge plants contained relatively high Se levels in all organs analyzed. However, the slightest Se variation among reproductive organs and leaves was observed in the Pine Ridge population, where the reproductive organs showed, on average, only 1.5-fold higher Se concentration when compared to the leaves.

Overall, the silique Se concentration showed a positive correlation with the Se levels found in the seeds for Coyote Ridge (Figure 4A) (R = 0.5522, *p* = 0.0266) and Pine Ridge plants (Figure 4B) (R = 0.7257, *p* = 0.0416). Another positive correlation (R = 0.5602, *p* = 0.0240) was found between leaf Se and seed Se for the Coyote Ridge plants (Figure 4E).

### 2.2. Herbivory and Fitness Parameters

Further analyses aimed to understand if the leaf Se concentration of *S. pinnata* plants correlated with reduced herbivory percentage, owing to the ecological benefit of reduced herbivory attack. To better understand our results, it is essential to remember that the Coyote Ridge samples were collected in the Fall (September), while the Pine Ridge samples were collected during the Spring (May), when the herbivory damage is expected to be lower and leaf Se concentration is higher [41]. An indication of a possible positive correlation was found between the percentage of herbivory and the leaf Se accumulation of Coyote Ridge plants (R = 0.4775, *p* = 0.0526) (Figure 5C), which could be attributed to the relatively low Se levels and the activity of Se tolerant herbivores. However, no evident correlation was found in the Pine Ridge plants. A significant (2-fold) difference in the number of leaves per plant was found between the two areas analyzed. On average, the *S. pinnata* plants from Coyote Ridge had 146 leaves (Figure 5A), while Pine Ridge plants had 81 leaves (Figure 5D). The percentage of leaves with signs of herbivory damage per plant was about fivefold higher in Coyote Ridge, where the plants showed on average more than 85% of herbivory (Figure 5B). In contrast, on average, *S. pinnata* plants grown in Pine Ridge showed only 18% of herbivory (Figure 5E).

Overall, no correlation was found between the number of siliques per plant and silique Se. The Coyote Ridge plants had fewer siliques and seeds than those collected at the Pine Ridge site. The average number of siliques per plant was 83 for Coyote Ridge (Figure 6A) and 469 for Pine ridge plants on average (Figure 7A), which corresponds to a 5-fold difference. When the seeds were analyzed, no correlation was found between the source Se and the number of seeds per plant (Figure 6D and Figure 7D) for both sites. A moderate positive correlation (R = 0.4895, *p* = 0.0543) between the average seed weight per plant and seed Se was found for the Coyote Ridge plants (Figure 6F). However, no significant correlation was found for the Pine Ridge plants. The average number of seeds per plant between the two studied sites was more than 30-fold different; the Coyote Ridge plants showed an average of 143 seeds per plant (Figure 6C), in contrast to the average of 2000 seeds found the Pine Ridge plants (Figure 7C). The average seed weight per plant did not significantly differ between Coyote Ridge and Pine Ridge plants. The average seed weight per plant in Coyote Ridge was 1.4 mg (Figure 6E) and 1.5 mg in Pine Ridge (Figure 7E).

### 2.3. Total Leaf Phenolics and Antioxidant Capacity

The antioxidant capacity, expressed in Trolox (vitamin E equivalents) antioxidant capacity (TEAC), and the amount of leaf total phenolics in terms of gallic acid equivalents (GAE, Appendix A) were analyzed to investigate further the effect of Se hyperaccumulation in *S. pinnata* plants. Overall, the *S. pinnata* plants showed higher total antioxidant capacity when growing at Coyote Ridge than those plants growing at Pine Ridge. The average concentration of vitamin E equivalents found in Coyote Ridge plants was 299 µmol g^−1^ DW of TEAC (Figure 8), and in Pine Ridge, the number was 1.5-fold lower, at 175.63 µmol g^−1^ DW (Figure 9). A significant negative correlation (R = −0.0743, *p* = 0.0106) between the average seed weight and the total antioxidant capacity was found for the Coyote Ridge plants. However, no significant difference in total leaf phenolics was found between plants from both sites. The average leaf phenolics in Coyote Ridge was 28.54 mg g^−1^ DW of GAE (Appendix A), while in Pine Ridge, this number was 27.5 mg g^−1^ DW (Appendix A). No significant correlation was found among the average leaf phenolics and leaf Se (Appendix A), percentage of herbivory (Appendix A), or average seed weight for both sites (Appendix A).

### 2.4. Glucosinolates Quantification

The last set of analyses aimed to understand if the high levels of Se would affect the concentration of glucosinolates (GLSs), which are S-containing compounds, due to Se and S antagonism for the uptake and assimilation. Moreover, GLSs play critical ecological roles in plants. Progoitrin was the most abundant GLS identified in both leaves and seeds of *S. pinnata*. Our data show no evidence that high Se levels affect the GLS concentration. Furthermore, no negative or positive statistically significant correlation was found between Se and GLS in leaves or seeds of *S. pinnata* (Figure 10 and Figure 11).

## 3. Discussion

This research aimed to analyze to what extent the variation in Se accumulation can affect hyperaccumulator *S. pinnata* fitness, as judged from different parameters for physiological and biochemical performance and herbivory while growing in two seasons on two seleniferous sites, Coyote Ridge and Pine Ridge Natural Areas. While there was substantial variation in Se concentration within each population, most plants within each population had high Se concentration, especially in the reproductive parts. No evidence for positive or negative correlation was found between leaf, silique, or seed Se concentration with any of the fitness or biochemical parameters number of leaves per plant, degree of leaf herbivory, number of siliques and seeds per plant, average seed weight, total leaf phenolics, and total leaf glucosinolates. The lack of correlation between the level of Se concentration and apparent fitness indicates that the observed variation in Se has no effects. However, it may be that protective effects of Se against biotic stresses or via enhanced antioxidant capacity are already pronounced at low Se tissue levels and that these effects saturate at a particular tissue Se threshold. Most of the plants examined here may have Se levels above this threshold. While there was no indication of Se-conferred enhanced *S. pinnata* fitness within the tissue Se range observed, there was also no indication for fitness being compromised by Se hyperaccumulation due to, for example, toxicity to the plant itself.

In both populations, there was a clear difference between vegetative and reproductive organs with respect to the degree of variation in tissue Se levels: leaf Se levels varied 55-fold in the Coyote Ridge population and 13-fold in the Pine Ridge population. In contrast, the seeds showed consistently high Se levels, which varied only 4-fold and 3-fold in Coyote Ridge and Pine Ridge populations, respectively. The Se levels found in leaves were not all at hyperaccumulator level (some were below the threshold of 1000 mg kg^−1^ DW). In contrast, all seeds and siliques samples analyzed from both sites showed hyperaccumulator Se levels.

The observed variation in overall plant Se accumulation may be caused in part by local variation in soil Se concentration and bioavailability (pH, organic matter, and microbial composition) [12,42] and in part by variation within populations in expression levels of genes related to hyperaccumulation, such as those involved in sulfate/selenate transport and assimilation [20]. The physicochemical properties of Pine Ridge soils were investigated [29], where a relatively high organic matter and slightly basic soil were reported. Furthermore, some variation in soil Se concentration and distribution in the soil was previously observed. The described soil Se levels in Pine Ridge natural area range from 2 to 23 μg g^−1^ DW, where the highest Se levels are reported to be present in areas where Se hyperaccumulator plants are present. The reported soil Se levels in Coyote Ridge is relatively lower, ranging from 0.9 to 2.2 μg g^−1^ DW, and interestingly, the highest Se levels were found in plots where no hyperaccumulator plants were observed [13]. Regardless of the underlying cause(s), the resulting variation in tissue Se concentration apparently did not affect any of the performance parameters analyzed. Still, evolutionary fitness could only be affected if there is an underlying genetic cause.

The finding that leaf Se was more variable than seed Se may be explained by earlier observations. *Stanleya pinnata* was shown to redistribute Se specifically (independently from S) via remobilization from aging leaves, resulting in Se levels that are highest in young leaves and the pollen and ovules of flowers, as well as in seeds [15,19,37]. *S. pinnata* may preferentially store Se in its reproductive organs because it offers seed and seedling protection from biotic and abiotic stresses [37]. Within young leaves, Se is sequestered in the vacuoles of epidermal cells at the edge of the leaf, achieving maximal herbivory and pathogen protection [19,35]. These Se partitioning preferences may confer plant fitness, associated with a plant’s productivity and reproductive success. From a practical perspective, the more consistently high levels of Se in the reproductive organs indicate that these organs are a very reliable indicator of hyperaccumulator status and can be sampled in addition to leaves when available.

Our data show no negative correlation between plant Se concentration and herbivory damage. There was a weak positive correlation between leaf Se accumulation and leaf herbivory for Coyote Ridge plants (R = 0.4775, and *p* = 0.0526; see Figure 5C). The Coyote Ridge samples were collected in September, when seasonal variation in leaf Se concentration is lowest [41], explaining the almost 3-fold lower average Se concentration than the Pine Ridge samples collected in May, explaining their overall higher degree of herbivory damage. The relatively low Se levels in those leaves may have made them more edible for herbivores, especially those with elevated Se-resistance. Previous research has shown strong evidence that plant Se accumulation offers protection against a wide variety of different generalist herbivore species through deterrence and toxicity; these include aphids [43], moth and butterfly larvae [44], grasshoppers [26,28], thrips and spider mites [24], and prairie dogs [25]. However, there is also clear evidence of Se-resistant herbivores occurring in seleniferous areas that feed on hyperaccumulator leaves and seeds [32,33,34]. The new results presented here indicate that these Se-resistant herbivores pose a significant herbivory pressure for *S. pinnata*, considering that most leaves showed herbivory damage, even when the Se tissue levels were high. For instance, 31% of the Coyote Ridge plants showed leaf Se levels above the hyperaccumulator threshold, ranging from 1018 to 2565 mg Se kg^−1^ DW (Figure 5A). Many of these plants showed significant herbivory damage (Figure 5B). The presence and significant pressure of Se-resistant herbivore populations may obscure any protective effects of the accumulated plant Se against generalist herbivores, offering another reason for the lack of correlation between plant Se concentration and herbivory damage besides the explanation that most plants had Se levels above a typically protective threshold.

The effect of accumulation of high Se levels in plants goes beyond ecology. Ultimately, toxic levels of Se in tissues could directly impact plant physiology and reproduction. We found no evidence of such a constraint. Plant size in this study did not correlate with leaf Se concentration. The reproductive fitness of the studied *S. pinnata* plants, as estimated from the number of siliques and seeds and the average seed weight per plant, was not compromised by high Se levels.

On the contrary, a moderate positive correlation was observed between the average seed weight per plant and seed Se for the Coyote Ridge plants (Figure 6F). A beneficial effect of Se fertilization on plant productivity is well documented at low Se levels, e.g., mustard seeds [45] and lentil seeds [46,47]. Selenium can exert positive physiological effects at low concentrations that could result in higher seed quality, such as improving overall growth and development enhancing photosynthesis, resulting in higher accumulation of starch and sugars [5,48].

It has been reported that Se increases the transcript levels and activity of different antioxidant enzymes, thereby regulating the concentration of ROS and overall tissue antioxidant response [3]. Hyperaccumulator species even seem to benefit from Se at tissue levels that are toxic to non-hyperaccumulator species. For example, when Se was supplied at hyperaccumulator levels to *S. pinnata*, the overall growth and reproductive parameters, such as pollen tube growth, were improved [29,35,37]. The new field data from this study agree with these lab studies and indicate that the capacity for Se tolerance does not appear to be a physiological constraint for Se hyperaccumulation in *S. pinnata* in the field.

To analyze how tissue Se concentration influences the antioxidant activity in leaves of *S. pinnata* plants grown in the field, the total antioxidant capacity and the levels of leaf total phenolics were investigated. Under a certain threshold, a relatively higher yield of polyphenols in the extract can indicate a higher cellular antioxidant activity. However, no correlation between the total antioxidant capacity and the total leaf Se or the percentage of herbivory was found. Furthermore, the Coyote Ridge samples showed, on average, a higher antioxidant capacity than those from Pine Ridge, which could be explained by an elevated concentration of antioxidant compounds due to the higher state of herbivory of those plants, as described before. Interestingly, a significant negative correlation between the average seed weight and the total antioxidant capacity was found for the Coyote Ridge plants. Oxidative stress is characterized by an imbalance in the cellular redox state due to the overproduction of reactive oxygen species (ROS) above the cellular antioxidant capacity [49,50,51]. Even though ROSs are known to have significant signaling roles in seed germination [52], their overproduction can lead to the destruction of lipids, proteins, and nucleic acids, resulting in impaired development and overall lower seed weight. The cellular enzymatic and non-enzymatic antioxidant defense mechanisms can directly neutralize the excessive levels of ROS. Therefore, if these seeds are experiencing some level of oxidative stress, it is expected that there would be a higher production of different antioxidant enzymes and secondary metabolites, but this was not found here.

Furthermore, no correlation was found between the average of the total leaf phenolics and all the other analyzed parameters (total leaf Se, percentage of herbivory, and average seed weight). It seems counterintuitive that a higher Se concentration does not correspond with a higher antioxidant capacity. However, *S. pinnata* plants also have an elevated tissue concentration of sulfur (S) and S-containing metabolites (GSH and GSSG) because of the constitutively high expression of different sulfate transporters and S assimilation pathway enzymes [20]; this might result in a higher reactive oxygen species (ROS) scavenging capacity and a better response to oxidative stress, independent of the Se-status of the plant. Furthermore, Se at low levels improves the cellular antioxidant response in plants [3,5]. Perhaps the correlation between the Se and antioxidant capacity might be more apparent at a lower Se concentration range or possibly lowered antioxidant capacity when Se is lacking.

Plants utilize different mechanisms to defend their tissues from herbivory and pathogens. Accumulating particular toxic elements from the environment as an elemental defense strategy is generally considered to be relatively cost-efficient [22,53], as compared to other strategies by which some species utilize more energy-costly physiological strategies to cope with biotic stress. The central defense secondary metabolites in the Brassicaceae family are the glucosinolates (GLSs), a large group of sulfur and nitrogen-containing compounds responsible for herbivory protection and other ecological roles. The enzyme myrosinase initiates the hydrolysis of the GLS into its active forms, and it is stored in different cells from the GLS. When herbivores damage the tissues, the myrosinase comes into contact with its substrate, forming glucose and the unstable aglycone, which is later converted to the active compounds [54,55].

*Stanleya pinnata* was found here to contain high concentrations of GLS compounds, in addition to its high Se levels. Thus, it seems these plants are using two different mechanisms for herbivory protection. Our study did not find any correlation between Se and GLS in leaves or seeds of *S. pinnata*, indicating the GLS metabolism is not affected by Se in the HA. Since GLS are S-containing metabolites, higher levels of Se could negatively impact the GLS pool in the tissues. Such inhibition was already reported by Tian et al. [56], where the supplementation with 25 µM of sodium selenate lowered the expression of several genes in the GLS biosynthesis pathway and significantly reduced the concentration of the GLS-precursor amino acids methionine and phenylalanine, and the GLS concentration in leaves and florets of broccoli, without affecting the S status of the plant. Similar results were reported by Toler et al. [57], where the concentrations of different GLSs were reduced in the presence of selenate in *Brassica oleracea*, even under regular S supplementation. It appears that the hyperaccumulator *S. pinnata* differs from its Brassicaceae relatives in this respect, as it does in other aspects of Se-S interactions.

Other studies reported the presence of SeGLS in Brassica spp. (secondary Se accumulators), specifically the family of glucosinolates containing the methylthio (CH_3_–S–) group (MeS-GLS) [54]. Matich et al. [54] found SeGLS in broccoli florets and leaves as glucoselenoerucin, cauliflower florets, and stems. SeGLS was found as glucoselenoiberverin [55], and in forage rape taproots as glucoselenoerucin [55], after 5.0 mM sodium selenate supplementation for four weeks. It has been suggested that, at low concentrations, Se can be incorporated into the methylthioalkyl moiety of GLS from the amino acid SeMet in the Brassica spp., replacing its S analog without disrupting the formation of S-GLS [55]. To date, the presence of SeGLS in *S. pinnata* has not yet been reported. Our finding that GLS metabolism is not affected by high levels of Se in this Se-hyperaccumulator may suggest a mechanism for excluding Se from these compounds.

## 4. Materials and Methods

### 4.1. Plant Material, Study Sites, and GPS Coordinates

Biological materials of different *Stanleya pinnata* L. (Brassicaceae) plants were collected at two different sites: Coyote Ridge natural area (geographic coordinates: latitude, 40°28′51″ N; longitude, 105°07′31″ W), and Pine Ridge natural area (geographic coordinates: latitude, 40°32′32″ N; longitude, 105°08′04″ W). Both natural areas have been described before [13,14] and are located on a seleniferous formation West of Fort Collins in the state of Colorado, in the United States of America. Samples from 23 different plants were collected in Pine Ridge natural area, and a total of 24 individual plants from the Pine Ridge natural were used.

A total of seven field trips were conducted to the study sites during 2017, 2018, and 2019. In 2017, two trips to each location were made to collect leaves (branches), seeds, and siliques. Pine Ridge: May (Spring) and September (Fall). Coyote Ridge: July (Summer) and September (Fall). In 2018, one trip was made to each site to collect leaves (branches). Pine Ridge: August (Summer). Coyote Ridge: August (Summer). On May 2019 (Spring), one last sampling trip was made to Pine Ridge to collect leaf samples.

The GPS coordinates for each plant were recorded by using a Garmin Oregon 650t GPS. The GPS points were managed by using EasyGPS (version 6.11 TopoGrafix Edition), and the satellite images shown were generated by using Google Earth (Version 9.129.0.1).

### 4.2. Determination of Selenium Concentration

After drying at 50 °C until constant weight, 100 mg of *S. pinnata* seed and leaf samples from each study site was weighed for elemental analysis. These samples were then digested with 1 mL of nitric acid as follows [58]: the samples were heated for two hours at 60 °C and six hours at 125 °C, and then diluted to 10 mL with distilled water. Inductively coupled plasma–optical emission spectroscopy (ICP–OES) was used to analyze the digested seed samples’ elemental composition.

### 4.3. Herbivory and Fitness Parameters

The following fitness parameters were analyzed: total number of leaves per plant, number of siliques per plant, number of seeds per plant, and average individual seed weight. The total number of leaves with herbivory, number of intact leaves, and percentage of leaves with herbivory were also analyzed per plant. The total number of branches per plant in the field was recorded to help estimate the mentioned parameters. One to three branches per plant were collected and brought to the lab, and the number of leaves and siliques per branch was counted. The numbers per branch were then multiplied by the total number of branches per plant in the field to estimate the total number of leaves and siliques per plant.

All the siliques from the same plant were opened, and the seeds were collected in one microcentrifuge tube. Ten random seeds from the same plant were then weighed by using a precision scale (Mettler Toledo, AB204-S/FACT). This number was then divided by 10 to estimate the average seed weight for that plant.

### 4.4. Total Leaf Phenolics and Antioxidant Capacity

Leaf samples were lyophilized, powdered, and weighed. The freeze-dried material was extracted with 80% acetone at a ratio of 25 µL/1 mg tissue while rotated in the dark at 4 °C for 30 min. The supernatant was collected, diluted with additional acetone at 1:10 or 1:20 depending on the sample, and stored on ice until used. All samples for this assay were read at 734λ, using a PowerWaveXS2 UV–vis spectrophotometer (BioTek Instruments, Winooski, VT, USA), using the method of Miller and Rice-Evans [59]. Trolox (Vitamin E equivalents) was the standard used for this assay, and the results are expressed as micromoles of Trolox-equivalent antioxidant capacity (TEAC) per gram dry weight (µmol g^−1^ DW).

Diluted supernatant collected from the extraction described above was used for total phenolics. Folin–Ciocâlteu (Sigma Chemicals, St. Louis, MO, USA) was used as described by Singleton and Rossi [60]. All samples for this assay were read at 765λ, using a PowerWaveXS2 UV–vis spectrophotometer, using gallic acid as a standard with results expressed as milligrams of gallic acid equivalent (GAE) per gram of dry weight (mg g^−1^ DW).

### 4.5. Glucosinolates Extraction and Quantification

Total glucosinolates were extracted from leaf and seed samples according to Argentieri et al. [61], but with some modifications [62]. Seeds (30 mg) were first frozen in liquid nitrogen and then grounded. GLSs were extracted by boiling the crushed seeds and leaf samples in an 18 mL methanol/water mixture (70:30, *v*/*v*) for 10 min to deactivate myrosinase. The supernatants were then dried (two extracts per sample) and resuspended in 500 mL methanol. Sinigrin (1.26 mg/mL concentration) was added to the solution as an internal standard. After 4 min, the solution was filtered at 0.45 μM (Millipore, Burlington, Massachusetts, US). The solutions were then boiled one more time in 70% methanol (*v*/*v*) for four minutes to ensure the complete extraction of total glucosinolates from the samples. The two extracts were further combined and purified once more by using a Solid-Phase Extraction (SPE) column (0.8 × 4 cm, Agilent Technologies, Cernusco sul Naviglio (Milan), Italy), equipped with 0.256 g of an ion exchange resin (DEAE-SE HADES-A25), imbedded in 4 mL of a 0.5 M Na-acetate buffer (pH = 5). The system was first washed with 1 mL of deionized water and then loaded with 2.5 mL of the extracted samples containing the standard. The column was further treated overnight with the enzyme sulfatase (41.6 mg/mL dilution) extracted from Helix pomatia-Type 1 (Roman snail) to convert the glucosinolates into the corresponding desulfated derivates. These derivates were further eluted from the column, using 2 mL of deionized water. Glucoerucin was used as a reference standard at different concentration levels to quantify glucosinolates.

### 4.6. Statistical Analysis

The software JMP Pro 15.0.0 (SAS Institute, Cary, NC, USA) was used for statistical data analysis. Multivariate analysis was used to individually compare the selenium concentration in different tissues and all the fitness parameters. Pairwise correlations were performed for each combination of variables. The correlation coefficient R and the *p*-values are shown in the scatterplots.

## 5. Conclusions

It was hypothesized that plant Se concentration positively correlates with the various fitness and physiological parameters, particularly at lower Se tissue levels, perhaps saturating at a certain tissue Se threshold. Moreover, an inverse correlation between Se concentration and herbivory is predicted. The field studies presented here suggest that the observed variation in Se hyperaccumulation does not enhance or compromise *S. pinnata* fitness when growing in its natural habitat on seleniferous soil. Despite the variation, most plants had high to very high Se levels, especially in reproductive organs. Plant size and reproductive parameters were not correlated with Se concentration, so the physiological capacity for Se tolerance does not appear to be a constraint for Se hyperaccumulation in this species. There was significant herbivory pressure even on the highest-Se plants, likely from Se-resistant invertebrate herbivores. Thus, while there may be Se-mediated herbivory protection to the hyperaccumulator from generalist herbivores, Se-resistant herbivores appeared to overcome this protective effect and are a significant presence in this native seleniferous habitat, perhaps limiting the ecological advantage of Se hyperaccumulation.

## Figures and Tables

**Figure 1 plants-11-00690-f001:**
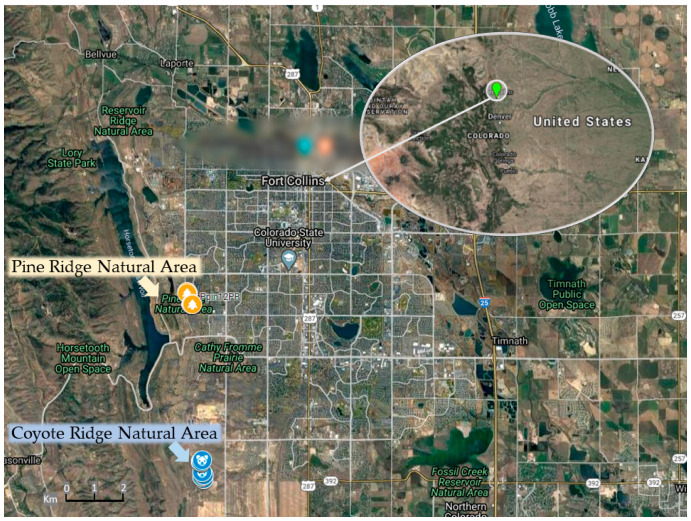
Map depicting the two natural areas investigated, namely the Coyote Ridge natural area (geographic coordinates: latitude, 40°28′51″ N; longitude, 105°07′31″ W) and the Pine Ridge Natural area (geographic coordinates: latitude, 40°32′32″ N; longitude, 105°08′05″ W), near the city of Fort Collins, Colorado/USA.

**Figure 2 plants-11-00690-f002:**
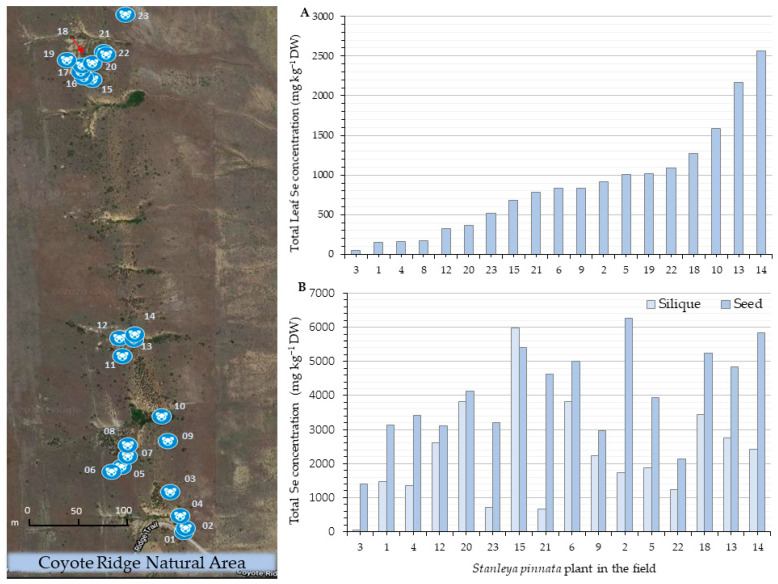
*Stanleya pinnata* plant locations at the Coyote Ridge natural area (**left**) and tissue Se concentrations (**right**). The numbers on the horizontal axis represent the collection numbers of each plant. (**A**) Leaf Se concentration of individual plants, ordered according to their Se levels. (**B**) Silique and seed Se concentration of the same plants, where the numbering was maintained for consistency.

**Figure 3 plants-11-00690-f003:**
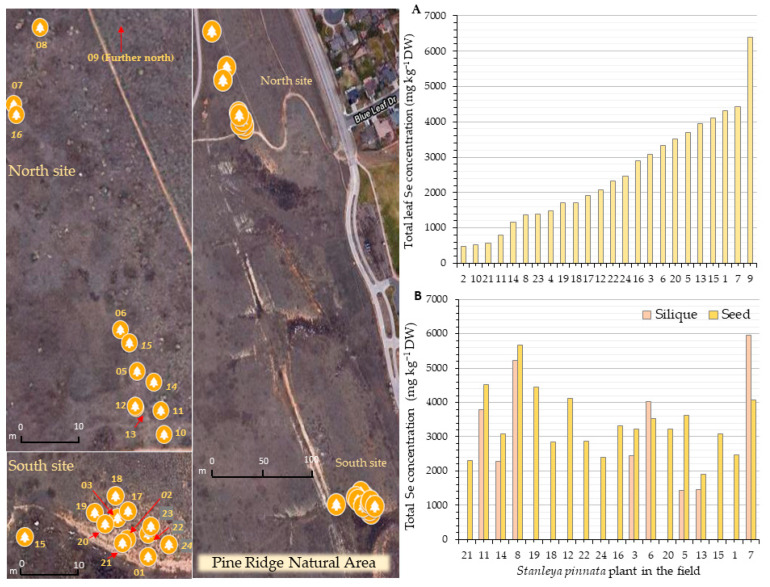
*Stanleya pinnata* plant locations at the Pine Ridge natural area (**left**) and tissue Se concentrations (**right**). The numbers on the horizontal axis represent the collection number for each plant. (**A**) Leaf Se concentration ordered according to their Se levels. (**B**) Silique and seed Se concentration for the same plants; the numbering was maintained for consistency.

**Figure 4 plants-11-00690-f004:**
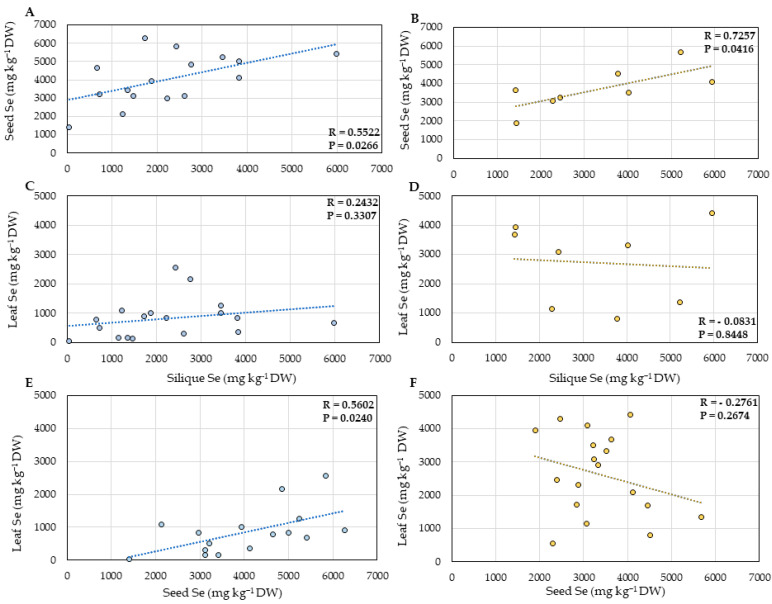
Correlation between seed and silique Se concentration (**A**,**B**), leaf and silique Se concentration (**C**,**D**), and leaf and seed Se concentration (**E**,**F**). Correlation coefficient R and *p*-value are shown in each panel. Panels (**A**,**C**,**E**) show the Coyote Ridge data. Panels (**B**,**D**,**F**) show the Pine Ridge data.

**Figure 5 plants-11-00690-f005:**
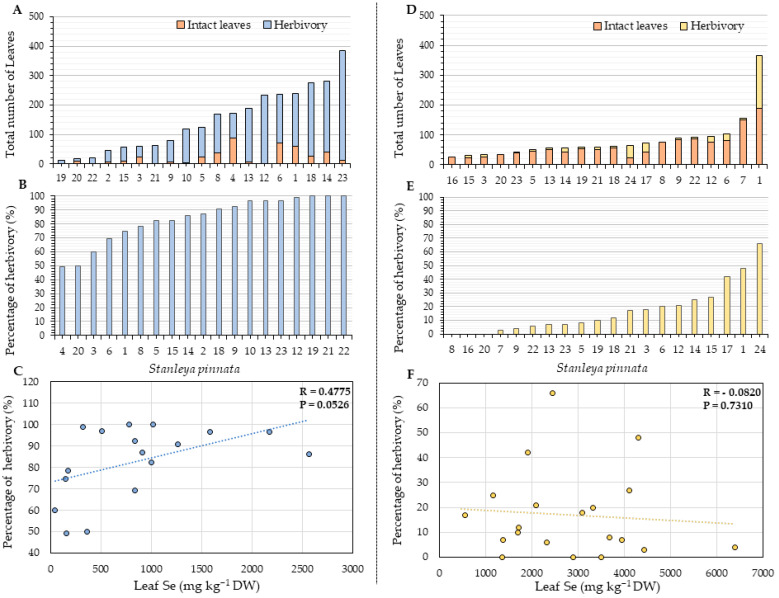
Total number of leaves with and without signs of herbivory per plant (**A**,**D**). The state of herbivory is represented as the percentage of leaves with herbivory per plant (**B**,**E**). Correlation between leaf Se concentration with herbivory (**C**,**F**) for *S. pinnata* plants growing at Coyote Ridge (**A**–**C**) and Pine Ridge (**D**–**F**). The numbers on the horizontal axis (**A**,**B**,**D**,**E**) represent the collection numbers of individual plants at the locations indicated in Figure 2 and Figure 3. The graph shows the correlation coefficient R and *p*-value for panels (**C**,**F**). Coyote Ridge samples were collected in September Pine Ridge samples in May when herbivory damage was lower.

**Figure 6 plants-11-00690-f006:**
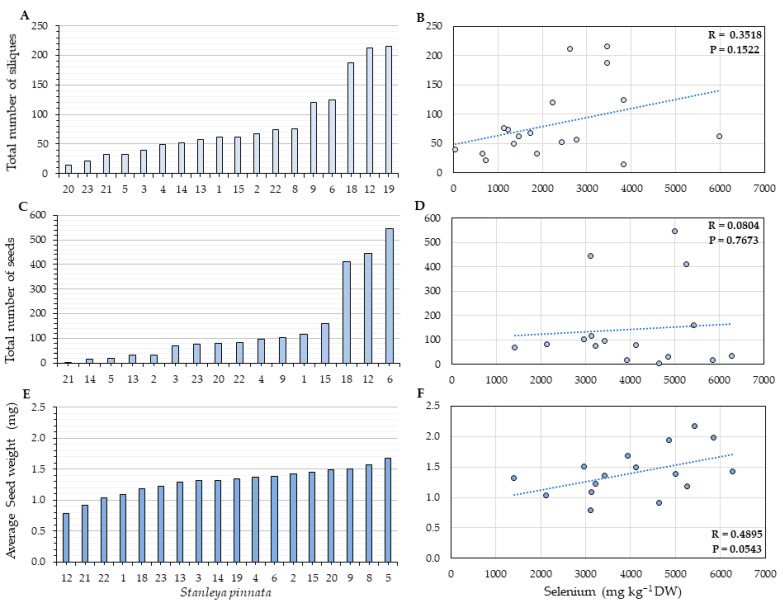
Reproductive fitness parameters (**A**,**C**,**E**) and their correlation with silique (**B**) or seed (**D**,**F**) Se concentration for *S. pinnata* plants growing at Coyote Ridge. (**A**,**B**) Total number of siliques per plant, (**C**,**D**) total number of seeds per plant, and (**E**,**F**) average seed weight. The numbers on the horizontal axis (**A**,**C**,**E**) represent the collection numbers of individual plants at the locations indicated in Figure 2. The graphs show the correlation coefficient R and *p*-value for panels (**B**,**D**,**F**).

**Figure 7 plants-11-00690-f007:**
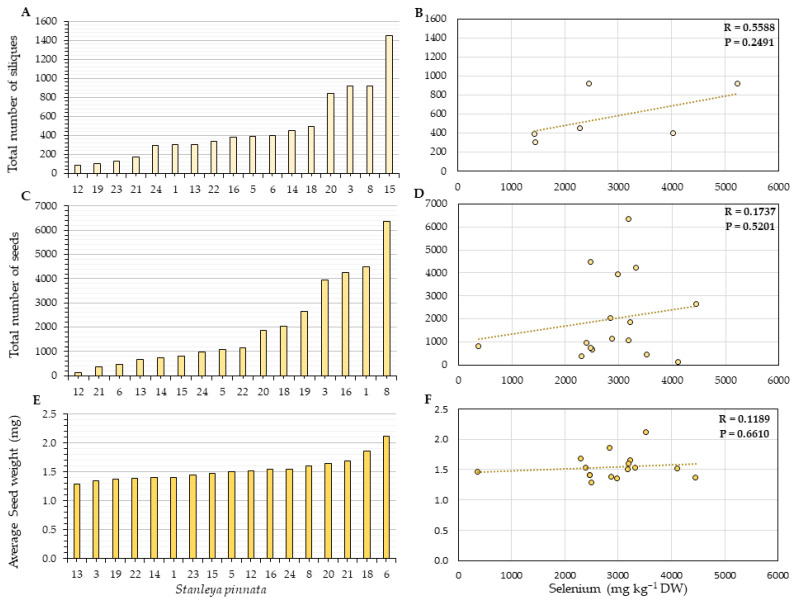
Reproductive fitness parameters (**A**,**C**,**E**) and their correlation with silique (**B**) or seed (**D**,**F**) Se concentration for *S. pinnata* plants growing at Pine Ridge. (**A**,**B**) Total number of siliques per plant, (**C**,**D**) total number of seeds per plant, and (**E**,**F**) average seed weight. The numbers on the horizontal axis (**A**,**C**,**E**) represent the collection numbers for individual plants at the locations indicated in Figure 3. The graphs show the correlation coefficient R and *p*-value for panels (**B**,**D**,**F**).

**Figure 8 plants-11-00690-f008:**
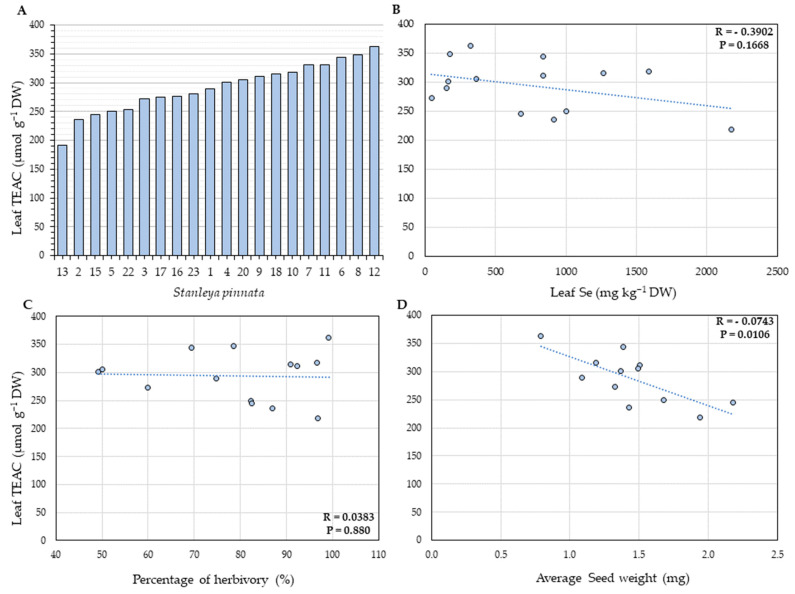
(**A**) Total antioxidant capacity in leaves of *S. pinnata* plants (Coyote Ridge), expressed as Trolox (Vitamin E equivalents) antioxidant capacity (TEAC). The error bars in panel (**A**) represent the standard deviation of the mean from three technical replicates per plant. (**B**–**D**) Correlation between leaf TEAC (vertical axis) with leaf Se concentration (**B**), percentage of herbivory (**C**), and average seed weight (**D**). The numbers on the horizontal axis (**A**) represent the collection numbers for individual plants at the locations indicated in Figure 2. The graphs show the correlation coefficient R and *p*-value for panels (**B**–**D**).

**Figure 9 plants-11-00690-f009:**
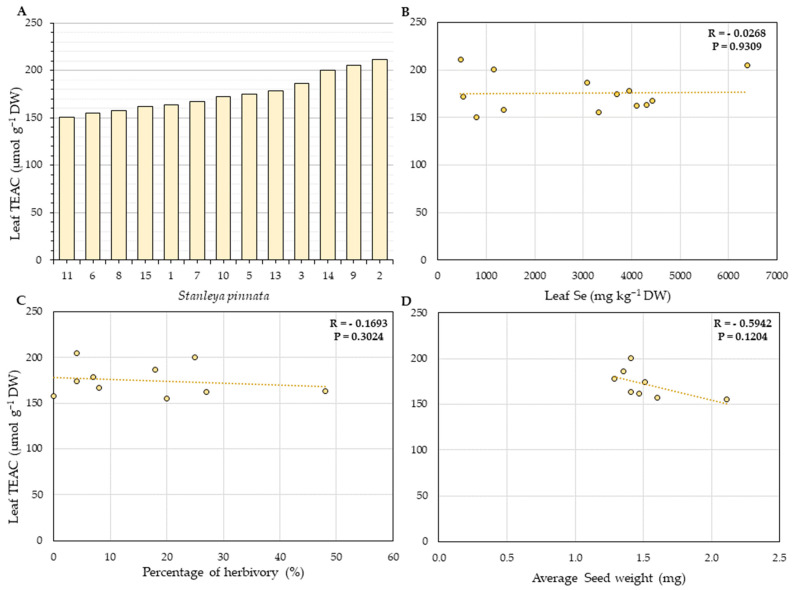
(**A**) Total antioxidant capacity in leaves of *S. pinnata* plants (Pine Ridge), expressed as Trolox (Vitamin E equivalents) antioxidant capacity (TEAC). The error bars in panel (**A**) represent the standard deviation of the mean from three technical replicates per plant. (**B**–**D**) Correlation between leaf TEAC (vertical axis) with leaf Se concentration (**B**), percentage of herbivory (**C**), and average seed weight (**D**). The numbers on the horizontal axis (**A**) represent the collection numbers for individual plants at the locations indicated in Figure 3. The graphs show the correlation coefficient R and *p*-value for panels (**B**–**D**).

**Figure 10 plants-11-00690-f010:**
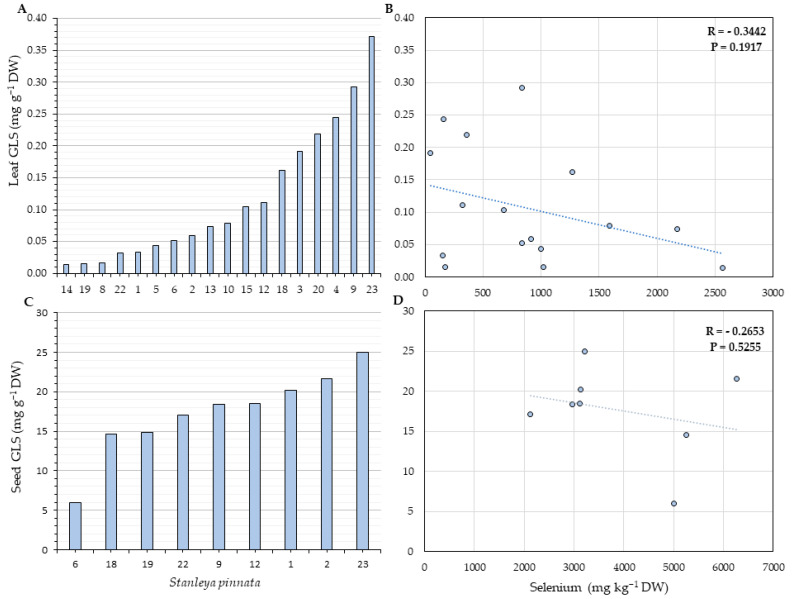
Total glucosinolates (GLSs) concentration in leaves (**A**) and seeds (**C**) and their correlation with leaf (**B**) and seed (**D**) Se concentration for *S. pinnata* plants growing at Coyote Ridge (locations shown in Figure 2). The numbers on the horizontal axis (**A**,**C**) represent the collection numbers for individual plants at the locations indicated in Figure 2. The graphs show the correlation coefficient R and *p*-value for panels (**B**,**D**).

**Figure 11 plants-11-00690-f011:**
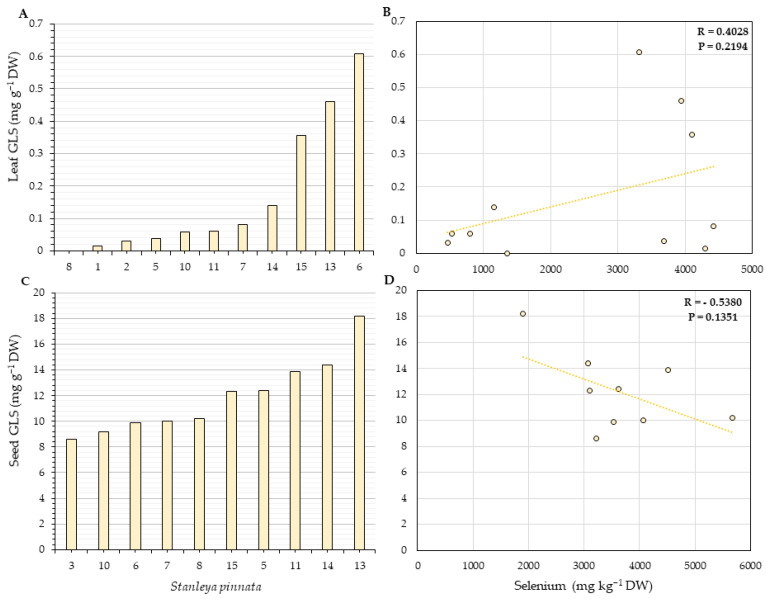
Total glucosinolates (GLSs) concentration in leaves (**A**) and seeds (**C**) and their correlation with leaf (**B**) and seed (**D**) Se concentration for *S. pinnata* plants growing at Pine Ridge (locations shown in Figure 3). The numbers on the horizontal axis (**A**,**C**) represent the collection numbers for individual plants at the locations indicated in Figure 3. The graphs show the correlation coefficient R and *p*-value for panels (**B**,**D**).

## Data Availability

Not applicable.

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
