# Peer review of "Hyperaccumulator Stanleya pinnata: In Situ Fitness in Relation to Tissue Selenium Concentration"

_plants, 2022, doi:10.3390/plants11050690_

Round 1
Reviewer 1 Report
A research was conducted to understand how variation in Se accumulation affects S. pinnata fitness from physiological and biochemical performance viewpoints.
Overall, a well-written and structured research. High in novelty and with a lot of exciting results. The only personal critique should be linked to the excessive paper length and the choice to present Results and Discussion separately, thus making reading less attractive.
Reviewer 2 Report
In this study, the authors analyzed the variation in Se accumulation and its effects on Stanleya pinnata fitness in two seleniferous sites, by studying physiological and biochemical parameters and herbivory. Overall the manuscript is well written. However, suggest a few changes to the authors before publication.
Line 23. I suggest rephrasing the sentence (starting with a digit does not seem proper).
Line 25. I suggest changing the units (mg Se/Kg and others as well) into a standard and more commonly used form (mg kg-1).
For figures 5 (A & B), 6 (A, B, & C), 7 (A, C, & E), 8(A & C), 9(A), 10 (A &C), 11 (A & C), I suggest to improve the presentation of number in the vertical axis. Only Stanleya pinnata does not provide a clear meaning. So I recommend, providing either the reference of figure 3 or explaining it individually in each graph (what the number stands for?)
Detailed soil physiochemical properties at the research sites should be provided to better understand the Stanleya pinnata fitness in the two experimental sites.
The authors should provide SD/SE values in Figures 2, 3, 5, 6, and 7, 10, & 11 (Same to Figure 8 and 9).
Reviewer 3 Report
The research is interesting and original. I have just a few comments.
- No research hypothesis is presented in the introduction;
-
I would advise describe in more detail the biology and ecology of Stanleya pinnata;
-
In the Figures No. 2 (A, B), 3 (A, B), 5(D, E, F), 6 (A, C, E), 7 (A, C, E), 10 (A, C), 11 (A, C) not presented statistical evaluation indicators;
- In the Figures No. 8 (A) and 9 (A) is not explained, what the vertical bars indicate;
-
No conclusions are presented in this article.
